# OpenReview forum: "Beyond One-Preference-for-All: Multi-Objective Direct Preference Optimization"
_ICLR.cc/2024/Conference — Submitted to ICLR 2024_

### Official Review · Reviewer_nwoD · 2023-10-29

**Soundness:** 3 good
**Presentation:** 3 good
**Contribution:** 2 fair
**Rating:** 5
**Confidence:** 3

**Summary:**

The paper examines the issue of fine-tuning large language models (LLMs) based on human preferences for multiple objectives. To align these objectives and optimize LLMs, the authors introduce MODPO, a method that does not rely on reinforcement learning (RL). Instead, it builds upon DPO [1] and trains using pairwise comparison data. The MODPO pipeline consists of two stages: reward modeling for each objective and preference optimization with a specific preference configuration (represented by a weight vector of rewards). Experimental results on safety alignment and long-form QA tasks show the effectiveness of MODPO.

[1] Rafailov R, Sharma A, Mitchell E, et al. Direct preference optimization: Your language model is secretly a reward model[J]. arXiv preprint arXiv:2305.18290, 2023.

**Strengths:**

1. Aligning LLMs with human preferences of multiple objectives is a promising research direction. To the best of my knowledge, this work represents one of the early efforts in this area.
2. The authors present comprehensive results to support the proposed methods.
3. The paper is easy to follow.

**Weaknesses:**

1. One of the greatest advantages of DPO is that it avoid reward modeling. MODPO, however, requires reward modeling to perform preference optimization. This makes me question if MODPO is the appropriate extension of DPO to the multi-objective setting.
2. If I understand correctly, DPO variants cannot be compared to MODPO, MORLHF, and Best-of-n in the experiments. This is because these variants only use pairwise comparison data, while MODPO utilizes both rewards and pairwise comparison data.
3. Building on the previous question, I am curious if MODPO will significantly underperform when rule-based rewards are unavailable. If rewards are learned from pairwise comparison data, will it outperform DPO LW in this scenario?

**Questions:**

1. In Figure 3, some data points of MODPO are far from the pareto frontier. Does it contradict with the claim that it "shows evidence of strong and reliable customizations"?

---

> ### Author Response · Authors · 2023-11-15
> **Response to Reviewer nwoD**
>
> Thank you for your thorough feedback and time spent reviewing our work. We really appreciate it.
>
> We respond to the questions / clarifications you raised below:
>
> > Q1: One of the greatest advantages of DPO is that it avoid reward modeling. MODPO, however, requires reward modeling to perform preference optimization. This makes me question if MODPO is the appropriate extension of DPO to the multi-objective setting.
>
> Ans1: Thanks for pointing out this question. Avoiding reward modeling is indeed one of the advantages of DPO. However, in terms of the more complex scenarios of multi-objective alignment, we find that our strategy of decoupling reward modeling and language modeling makes MODPO more broadly applicable and significantly outperforms other one-model-fits-all style extensions of DPO (e.g., DPO LW) (Figure 2). The discussion of how this decoupling allows MODPO for wider applicability is further elaborated in Ans2 and the discussion regarding how this decoupling helps MODPO outperform other extensions of DPO (e.g., DPO LW) is further elaborated in Ans3.
>
> > Q2: If I understand correctly, DPO variants cannot be compared to MODPO, MORLHF, and Best-of-n in the experiments. This is because these variants only use pairwise comparison data, while MODPO utilizes both rewards and pairwise comparison data.
>
> Ans2: Thank you for your feedback. We need to clarify that although it is true that MODPO utilizes both rewards and pairwise comparison data, **MODPO's flexibility in defining rewards means these** **rewards can also be learned from pairwise comparison datasets**. In contrast, other DPO variants (DPO soups / DPO LW) are much more restricted and only apply when all the datasets are collected in the form of pairwise comparison. **In other words, these DPO variants are not always applicable, but when the conditions for these DPO variants are satisfied, they can always be compared to MODPO(MORLHF/Best-of-n).** This is the case for our safety alignment experiments (Figure 2), where we consider two preference datasets and compare DPO variants to MODPO, MORLHF and Best-of-n. We will elaborate on the experiment details in Ans3.
>
> > Q3: Building on the previous question, I am curious if MODPO will significantly underperform when rule-based rewards are unavailable. If rewards are learned from pairwise comparison data, will it outperform DPO LW in this scenario.
>
> Ans3: Thank you for this insightful question. **In our safety alignment experiments, no rule-based rewards are available and rewards are actually learned from pairwise comparison data** (see experiment setups in section 4.1 and detailed implementation in Appendix D.2.1, updated paper)**.** Figure 2 shows that **MODPO significantly outperforms DPO LW:** the MODPO fronts consistently dominate the DPO LW fronts in both high KL and low KL regimes. This is because MODPO decouples the noise from two preference datasets by handling one objective at a time through multi-stage training whereas DPO LW concurrently learns two objectives from distinct noisy preference data, potentially impeding learning. In addition, **MODPO is also much more computationally efficient than DPO LW**. As discussed in the advantage part of the MODPO (section 3.2, updated paper), since the margin reward modeling (stage 1) of the MODPO can be greatly amortized, the per-LM training costs of MODPO are no more than that of DPO *on one preference dataset (*the margin term can be per-calculated and cached*).* However, the per-LM training costs for DPO LW amount to that of DPO *on a mixture of datasets.* This gap in computation efficiency is not negligible if we want to train a set of LMs instead of just one.
>
> > Q4. In Figure 3, some data points of MODPO are far from the pareto frontier. Does it contradict with the claim that it shows evidence of strong and reliable customizations?
>
> Ans4. Thank you for pointing this out. **No, it does not contradict with the claim.** Since we evaluate all LMs throughout training, 0.1 epoch until convergence, there must be some data points that stay close to where they start off (the SFT initialization). For example, if we focus on the $r_1^*$  vs $r_2^*$  quadrant in Figure 3, **the data points far from the fronts are actually evaluated in the middle of training, even at the very beginning of training**. The reason why we choose to visualize these points is that these checkpoints, although not optimal in terms of the obtained rewards, may be optimal in the rewards&KL trade-off. The KL with the SFT LM is also an important metric for fine-tuned LMs since obtaining slighter higher rewards with much larger KL is not necessarily ideal [1].
>
> References:
>
> [1] Rafael Rafailov, Archit Sharma, Eric Mitchell, Stefano Ermon, Christopher D. Manning, and Chelsea Finn. Direct preference optimization: Your language model is secretly a reward model, 2023.

---

> ### Author Response · Authors · 2023-11-20
> **Hope to Get Your Response**
>
> Dear reviewer nwoD,
>
> We would like to first thank you again for your constructive comments and helpful suggestions. Since we are nearly at the end of the discussion phase, we would like to post a follow-up discussion.
>
> In our previous response, we have clarified the raised questions and made corresponding improvements in the updated paper. We hope to further discuss with you whether your concerns have been addressed or not. If you still have any unclear parts of our work, please let us know.

---

> ### Author Response · Authors · 2023-11-22
> **Nearing End of Discussion Period**
>
> Dear reviewer nwoD,
>
> Hello! Since the discussion period is rapidly approaching to the end, we would like to kindly ask you to take a look at our responses and reevaluate our work given the clarifications we made. Please let us know whether our response addresses your concerns or whether there is any further detail we can provide to help address them. We really appreciate your time and consideration!

---

> ### Comment · Reviewer_nwoD · 2023-11-23
>
> Thank you for your comprehensive response.
>
> > MODPO is also much more computationally efficient than DPO LW.
>
> Could you provide more details such as the total number of GPU hours required for fine-tuning LLaMa-7b?

---

> ### Author Response · Authors · 2023-11-23
> **Response w.r.t. GPU hours**
>
> Thank you for your response.
>
> The total number of GPU hours of MODPO and MORLHF is shown in Table 1.
>
> The total number of GPU hours of MODPO and DPO LW is shown in the table below (we will reflect this new table in the Appendix soon).
>
> |  | safe alignment |
> | --- | --- |
> | MODPO | 4.0$\pm$0.1 |
> | DPO LW | 7.3$\pm$0.2 |
>
>
> The per-LM training costs of MODPO approximate that of DPO on **one preference dataset** (margin is pre-calculated and cached) while the per-LM training costs of DPO LW approximate that of DPO on **a mixture of (two) preference datasets**. Note that DPO LW does not apply in the long-form QA task.
>
> Please let us know whether our response addresses your concerns or whether there is any further detail we can provide to help address them.

---

### Official Review · Reviewer_GBhK · 2023-11-01

**Soundness:** 3 good
**Presentation:** 3 good
**Contribution:** 3 good
**Rating:** 6
**Confidence:** 3

**Summary:**

Language models aligned with average human preferences may not suit diverse preferences. Recent approaches collect multi-dimensional feedback and train models on distinct rewards per dimension to customize for preferences. Multi-objective reinforcement learning for this is unstable and inefficient. This paper presents Multi-Objective Direct Preference Optimization (MODPO), an RL-free algorithm extending Direct Preference Optimization to multiple alignment objectives. MODPO efficiently produces a Pareto-optimal set of customized language models for diverse preferences, demonstrated in safety alignment and question-answering tasks.

**Strengths:**

* MODOPO enables customized language models without needing complex, unstable multi-objective reinforcement learning. The integration of linear scalarization into preference modeling is a creative way to reuse standard pipelines.
* The paper is technically strong, with a formal problem definition, clear methodology, and mathematical proofs demonstrating that MODPO optimizes the desired objective. Experiments across two tasks verify that MODPO matches or exceeds prior methods in quality and efficiency.
* MODPO provides an efficient way to produce customized language models catering to diverse user preferences. This enhances accessibility and avoids a one-size-fits-all approach.

**Weaknesses:**

* The experiments, while showing compute savings, are still limited in scale and scope. Testing on more alignment objectives, model sizes, and tasks would strengthen claims of efficiency and customization ability. ablation studies could isolate benefits.
* The choice to focus on two objectives may undersimplify real-world diversity of preferences. Experiments with 3+ objectives could reveal whether MODPO scales effectively.

**Questions:**

* Could you provide more intuition on why the multi-stage training provides benefits like stability and efficiency? Some analysis or ablation study isolating the impact would strengthen this claim.
* For the safe response examples, could you discuss if the responses seem "too safe" - is there a way to steer generation away from trivial safest responses?
* What scaling limitations have you found in practice when increasing the number of objectives beyond 2? Experiments on 3+ objectives could reveal interesting trends.
* You mention MODPO could be used online by training on unlabeled data - could you elaborate on this idea more? Seems interesting for wider applicability.

---

> ### Author Response · Authors · 2023-11-15
> **Response to Reviewer GBhK (Part 1/2)**
>
> Thank you for reviewing our paper. We really appreciate your positive feedback and great suggestions on improving our paper.
>
> We respond to the questions / clarifications you raised below:
>
> > Q1: The experiments, while showing compute savings, are still limited in scale and scope. Testing on more alignment objectives, model sizes, and tasks would strengthen claims of efficiency and customization ability. ablation studies could isolate benefits.
>
> Ans1: Thank you for your great comments and suggestions - we have added a new section on scaling-up experiments (Appendix B, updated paper) with more alignment objectives (three objectives) on top of the long-form QA task. Figure 5 (Appendix B, updated paper) shows that MODPO scales pretty well, consistently outperforming MORLHF **in terms of customization ability.** In these scaling-up experiments, MODPO still saves **computing resources** in terms of GPU hours (see the table below).
>
> |  | QA(rel-fact-comp) | QA (rel-fact-pref) |
> | --- | --- | --- |
> | MODPO | 12.5$\pm$0.2 | 11.3$\pm$0.1 |
> | MORLHF | 32.0$\pm$1.8 | 35.1$\pm$0.6 |
>
> > Q2: The choice to focus on two objectives may undersimplify real-world diversity of preferences. Experiments with 3+ objectives could reveal whether MODPO scales effectively.
>
> Ans2: Thank you for your great suggestions about scaling MODPO - please see Ans1 above.
>
> > Q3: Could you provide more intuition on why the multi-stage training provides benefits like stability and efficiency? Some analysis or ablation study isolating the impact would strengthen this claim.
>
> Ans3: Thank you for this question - please check out the updated paper with a focus on the “MODPO outline” (section 3) and “MODPO advantages” (section 3.2). These sections exist in the original version of the paper but we slightly refactored the structure of the paper to make the analysis easier to follow. Our general arguments are built upon the observation that **multi-stage training allows MODPO to** **decouple the complexity** of the multi-objective settings with **minimal overheads** compared to the single-stage DPO in terms of both stability and efficiency:
>
> **Stability**: The training stability of DPO mainly comes from the fact that **DPO relies solely on the binary cross-entropy loss on a static preference dataset. This is also true for MODPO.** With the margin reward models pre-trained in stage 1, the LM training of MODPO can also be reduced to solving a binary classification problem on a static preference dataset in stage 2. The training dynamics of DPO loss (Eq. 6) and MODPO loss (Eq. 14) will not be significantly different. This is empirically supported in Appendix D.3, which shows similar learning dynamics for both DPO loss (Eq. 6) and MODPO loss (Eq. 14) in terms of monotonically increasing training accuracy.
>
> **Efficiency**. The computation efficiency of MODPO brought by multi-stage training is more straightforward. As illustrated in the “MODPO outline” (section 3), the margin reward modeling (stage 1) should only be performed once for all the LMs. Because we generally need to obtain a lot of LMs for diverse preferences, **the training costs of margin rewards modeling can be amortized and neglected**. Therefore, the bottleneck for training LMs is the language learning (stage 2). And in stage 2, if we have the margin $m_\phi$ per-calculated and cached along with the dataset, **the per-LM training costs will be exactly the same as the DPO**.
>
> > Q4: For the safe response examples, could you discuss if the responses seem "too safe" - is there a way to steer generation away from trivial safest responses?
>
> Ans4: Thank you for pointing out this question. If I understand correctly, responses being "too safe" refers to the situation that the LMs trivially refuse to respond to any questions even if the questions are not adversarial at all. In the context of MODPO and the multi-objective preference alignment paradigm in general, having responses being “too safe” may not be a failure mode because this may satisfy the preference of some specific groups. If one finds the current LM being “too safe”, one can choose another LM with more weighting on helpfulness from the Pareto front and see if this LM can satisfy the needs.

---

> ### Author Response · Authors · 2023-11-15
> **Response to Reviewer GBhK (Part 2/2)**
>
> > Q5: What scaling limitations have you found in practice when increasing the number of objectives beyond 2? Experiments on 3+ objectives could reveal interesting trends.
>
> Ans5: Thank you for this excellent question - as mentioned in Ans1, we have added a new section of scaling-up experiments (Appendix B) with three objectives and found that MODPO scales effectively. The reason why we do not further scale up is that 1) we do not find an appropriate human feedback dataset for LM finetuning with more than 3 feedback dimensions and 2) further scaling is straightforward for the MODPO algorithm but will pose a challenge to the frontier visualization. However, we believe experiments with 3 objectives already show some promising scaling trends of MODPO and help to verify MODPO’s empirical advantages in real-world scenarios.
>
> > Q6: You mention MODPO could be used online by training on unlabeled data - could you elaborate on this idea more? Seems interesting for wider applicability.
>
> Ans6: Thank you for this great question - please see the updated paper with a focus on Appendix A.3.1. The recipe is simple. Suppose we only have an SFT model $\pi_{\text{sft}}$, a generic reward model $r$ that we want to use to steer our LM $\pi_\theta$, and a prompt dataset $D$.
>
> 1. We can start with collecting response pairs from  $\pi_{\text{sft}}$ to the prompts in $x \sim D$ and label with random preference (random preferred / dispreferred label). This yields $D_{\text{rand}}$.
> 2. Now we have an SFT model $\pi_{\text{sft}}$, a generic reward model $r$, and a randomly labeled preference dataset $D_{\text{rand}}$.
> The conditions for MODPO are satisfied, then we can use MODPO loss on the random preference dataset with the generic reward as margin, i.e.,  $L_{\text{MODPO}}(\pi_\theta;r, \pi_{\text{sft}}, D_{\text{rand}})$ to steer language model $\pi_\theta$. Note that the implicit objective introduced by this randomly labeled preference is a constant (more formally, it is a function of the prompt) that does not affect the fine-tuned LMs. Essentially, the generic reward $r$ is the only steering force.
>
> This process can be optionally iterated in an online manner by collecting new response pairs from the new fine-tuned LMs $\pi_\theta$, labeling new response pairs with random preference $D_{\text{rand}}$, and continue with $L_{\text{MODPO}}(\pi_\theta;r, \pi_{\text{sft}}, D_{\text{rand}})$ to further improve LMs $\pi_\theta$.
>
> This approach is not fully explored in this paper but might be an interesting future direction.

---

> ### Author Response · Authors · 2023-11-20
> **Hope to Get Your Response**
>
> Dear reviewer GBhK,
>
> We would like to first thank you again for your constructive comments and helpful suggestions. Since we are nearly at the end of the discussion phase, we would like to post a follow-up discussion.
>
> In our previous response, we have clarified the raised questions and made corresponding improvements in the updated paper. We hope to further discuss with you whether your concerns have been addressed or not. If you still have any unclear parts of our work, please let us know.

---

### Official Review · Reviewer_c1KT · 2023-11-05

**Soundness:** 3 good
**Presentation:** 3 good
**Contribution:** 3 good
**Rating:** 10
**Confidence:** 4

**Summary:**

This paper present MODPO, a fine-tuning method for large language models without reinforcement learning.

The method suits for human-in-the-loop setting, where human preference has diversity and only pareto optimal solutions are available while no universal optimal solutions can be found. The RL-free algorithm extends Direct Preference Optimization (DPO) for multiple alignment objectives.

The purpose and exprimental results show out-performance by producing a Pareto-optimal set of LMs that cater for diverse preferences with much less computation resources compared with RL-based method.

**Strengths:**

- Overall this paper is well-written with clear motivation - to overcome the resource-intensive RL-based fine-tuning methods for LMs, a RL-free framework that can fit better to a real-world situation where resources are limited.

- Clear notations and detailed implementation and ablation examples in the Appendix. These add value to the communication to a wider community

- The impact of the proposed algorithm is convincing - by conducting KL-controlled experiments.

**Weaknesses:**

- An adoption of DPO for fine-tuning LMs results in convincing results, however, conditions for delivering such results, and requirements in collecting datasets.

**Questions:**

- In Figure 1, the proposed MODPO is positioned as a marginalized reward formulation for fine-tuning LMs. This is generally in line with the well-known advantage of discriminative learning techniques in leveraging small datasets. What are the pros and cons of adopting this technique to deliver the Pareto-optimal solution? Are there alternative techniques available?

---

> ### Author Response · Authors · 2023-11-15
> **Response to Reviewer c1KT**
>
> Thank you for reviewing our paper and the positive feedback. We are glad you find the proposed method convincing.
>
> We respond to the questions / clarifications you raised below:
>
> > Q1: however, conditions for delivering such results, and requirements in collecting datasets.
>
> Ans1: Thanks for pointing this out - please see the updated version of our paper. There are some conditions for MODPO, but most of these conditions can be easily satisfied. Collecting multi-dimensional feedback datasets is a standard practice in current RLHF work, even in early projects like InstructGPT [1]. Additionally, we've included Appendix A.3 to demonstrate MODPO's applicability in less typical scenarios, such as the absence of a preference dataset or addressing single-objective alignment problems. We hope these details can add value to the proposed methods and be useful to the broad community.
>
> > Q2: In Figure 1, the proposed MODPO is positioned as a marginalized reward formulation for fine-tuning LMs. This is generally in line with the well-known advantage of discriminative learning techniques in leveraging small datasets. What are the pros and cons of adopting this technique to deliver the Pareto-optimal solution? Are there alternative techniques available?
>
> Ans2: Thanks for your great comments about the connections between MODPO and other methods. As I am not an expert in *"discriminative learning techniques in leveraging small datasets"*, I may not be in a position to offer in-depth responses in this area. However, should you provide relevant references, I would be eager to explore and discuss this aspect further.
>
> References:
>
> [1] Long Ouyang et al. Training language models to follow instructions with human feedback, 2022.

---

> ### Author Response · Authors · 2023-11-20
> **Hope to Get Your Response**
>
> Dear reviewer c1KT,
>
> We would like to first thank you again for your constructive comments and helpful suggestions. Since we are nearly at the end of the discussion phase, we would like to post a follow-up discussion.
>
> In our previous response, we have clarified the raised questions and made corresponding improvements in the updated paper. We hope to further discuss with you whether your concerns have been addressed or not. If you still have any unclear parts of our work, please let us know.

---

### Official Review · Reviewer_TGqd · 2023-11-09

**Soundness:** 3 good
**Presentation:** 3 good
**Contribution:** 2 fair
**Rating:** 5
**Confidence:** 3

**Summary:**

This paper extends Direct Preference Optimization (DPO)  to Multi-Objective Direct Preference Optimization (MODPO) directly. MODPO trains different LMs to represent different collective reward models that combine all objectives with specific weightings. With a simple cross-entropy loss, the LMs optimized against the MODPO objective are analytically the exact solutions of the original MORLHF objective.

**Strengths:**

This paper extends DPO to MODPO directly for solving the multi-objective problems with LM.

**Weaknesses:**

1. The contribution is limited because it is just a direct extension of an existing method.
2. The experiment fails to show multiple objectives, i.e., no less than three objectives.
3. This paper fails to evaluate the proposed method on real-world datasets like the TripAdvisor Hotel Review Dataset, which naturally includes multiple feedback scores from multiple aspects.

**Questions:**

1. I suggest the authors add experiments on more than two objectives on real-world datasets like the TripAdvisor Hotel Review Dataset, which naturally includes multiple feedback scores from multiple aspects.

---

> ### Author Response · Authors · 2023-11-15
> **Response to Reviewer TGqd**
>
> Thank you for your thorough feedback and your suggestions for improving our paper. We really appreciate it.
>
> We respond to the questions / clarifications you raised below:
>
> > Q1: The contribution is limited because it is just a direct extension of an existing method.
>
> Ans1:  Thank you for your feedback. Although MODPO is conceptually simple, it consistently outperforms other extensions of DPO (Figure 2).  MODPO is also more broadly applicable than other DPO variants (e.g., DPO LW and DPO soups) since all of these variants work under a much stronger assumption than MODPO, thus inapplicable in our long-form QA experiments (section 4.1, baselines, updated paper). **More importantly, MODPO is the only extension of DPO that consistently outperforms MORLHF (the current go-to approach for multi-objective alignment [1,2,3]) (Figure 2,3,4) with much lower computation costs (Table 1).**
>
> Therefore, we believe that MODPO is the best choice among other potential extensions of DPO for an important research question (multi-objective alignment for diverse human values) and will be useful to the broad academic community.
>
> > Q2: The experiment fails to show multiple objectives, i.e., no less than three objectives.
>
> Ans2: Thank you for your great comments and suggestions - we have added a new section of scaling-up experiments (Appendix B) with three objectives on top of the long-form QA task. Figure 5 (Appendix B) shows that MODPO scales pretty well, consistently outperforming MORLHF!
>
> > Q3: This paper fails to evaluate the proposed method on real-world datasets like the TripAdvisor Hotel Review Dataset, which naturally includes multiple feedback scores from multiple aspects.
>
> Ans3: Thank you for your great comments and suggestions about real-world datasets. **However, the datasets we consider - BeaverTails [1] and QA-Feedback [2] are all real-world human feedback datasets from very recent works**. Our experiment settings are divided into two parts: one dealing with model-generated synthetic feedback and the other dealing with real human feedback (section 4.1, evaluation). The first synthetic setting is only for fast evaluation and we put an emphasis on the latter. Figure 3,4 show that for these two real-world datasets, MODPO consistently outperforms other baselines, including MORLHF. **See Ans4 for reasons why TripAdvisor Hotel Review Dataset is not the best fit for our experiments.**
>
> > Q4: I suggest the authors add experiments on more than two objectives on real-world datasets like the TripAdvisor Hotel Review Dataset, which naturally includes multiple feedback scores from multiple aspects.
>
> Ans4: Thank you for your valuable suggestions - as mentioned in Ans2, we have added more scaling-up experiments (Appendix B) with more alignment objectives (3) on top of the long-form QA and found pretty good scaling trends. We also appreciate your suggestion with the TripAdvisor Hotel Review Dataset, but the focus of our paper is to **finetune LM** from human feedback. **Since the TripAdvisor [4, 5] dataset primarily gathers feedback about hotels, rather than on LM outputs, and as generating hotel reviews is not practically useful for an LM assistant, this dataset does not align well with our experimental needs.**
>
> References:
>
> [1] Jiaming Ji, Mickel Liu, Juntao Dai, Xuehai Pan, Chi Zhang, Ce Bian, Chi Zhang, Ruiyang Sun, Yizhou Wang, and Yaodong Yang. Beavertails: Towards improved safety alignment of llm via a human-preference dataset, 2023.
>
> [2] Zeqiu Wu, Yushi Hu, Weijia Shi, Nouha Dziri, Alane Suhr, Prithviraj Ammanabrolu, Noah A Smith, Mari Ostendorf, and Hannaneh Hajishirzi. Fine-grained human feedback gives better rewards for language model training. arXiv preprint arXiv:2306.01693, 2023.
>
> [3] Alexandre Rame, Guillaume Couairon, Mustafa Shukor, Corentin Dancette, Jean-Baptiste Gaya, Laure Soulier, and Matthieu Cord. Rewarded soups: towards pareto-optimal alignment by interpolating weights fine-tuned on diverse rewards, 2023.
>
> [4] [https://notebook.community/melqkiades/yelp/notebooks/TripAdvisor-Datasets](https://notebook.community/melqkiades/yelp/notebooks/TripAdvisor-Datasets)
>
> [5] [https://www.cs.cmu.edu/~jiweil/html/hotel-review.html](https://www.cs.cmu.edu/~jiweil/html/hotel-review.html)

---

> ### Author Response · Authors · 2023-11-20
> **Hope to Get Your Response**
>
> Dear reviewer TGqd,
>
> We would like to first thank you again for your constructive comments and helpful suggestions. Since we are nearly at the end of the discussion phase, we would like to post a follow-up discussion.
>
> In our previous response, we have clarified the raised questions and made corresponding improvements in the updated paper. We hope to further discuss with you whether your concerns have been addressed or not. If you still have any unclear parts of our work, please let us know.

---

> ### Author Response · Authors · 2023-11-22
> **Nearing End of Discussion Period**
>
> Dear reviewer TGqd,
>
> Hello! Since the discussion period is rapidly approaching to the end, we would like to kindly ask you to take a look at our responses and reevaluate our work given our clarifications as well as the supplementary experiments. Please let us know whether our response addresses your concerns or whether there is any further detail we can provide to help address them. We really appreciate your time and consideration!

---

### Author Response · Authors · 2023-11-15
**General Response to All Reviewers**

We thank the reviewers for all the detailed comments and helpful suggestions. We have highlighted all the important changes in blue in the revised version of our paper (typos corrected are not highlighted). Here we provide an overview of our changes.

## Scaling-Up Experiments with 3+ Objectives (Appendix B)

We thank **Reviewer** **TGqd** and ****Reviewer GBhK**** for their suggestions on further scaling MODPO to 3+ objectives. We have added a new section of scaling-up experiments (Appendix B) with three objectives on top of the long-form QA task. **Figure 5 (Appendix B)** shows that MODPO consistently outperforms MORLHF for multiple combinations of objectives, demonstrating a promising scaling trend. The table below shows that MODPO also saves a lot of computation costs (in terms of GPU hours) compared to MORLHF for the two scaling-up experiments.

|  | QA(rel-fact-comp) | QA (rel-fact-pref) |
| --- | --- | --- |
| MODPO | 12.5$\pm$0.2 | 11.3$\pm$0.1 |
| MORLHF | 32.0$\pm$1.8 | 35.1$\pm$0.6 |

## Other Minor Changes

In addition to the major modifications metioned above, we have also made some relatively minor changes / refactors to enhance the paper’s clarity without introducing any new intellectual ideas / contents.

 1. In title, we add “for language models” to emphasize the main scope of our paper.
 2. In abstract & section 1 & Figure 1 caption, we disambiguate the use of "objective" in the context of "MODPO objective" versus "multi-objective alignment" to avoid confusion. The former refers to the loss for practical optimization while the latter means the dimension of human preferences. We make several paraphrases to avoid the terms "MODPO objective" or "MORLHF objective".

1. In section 2.2 (data collection), we add a discussion asserting that MODPO's data collection process is widely practiced, underscoring its broad applicability, in response to **Reviewer** ****c1KT’s**** question on MODPO’s conditions for delivering good results.
2. In section 2.2, we formally define the terms preference vector and preference space.
3. We divide section 3 into two subsections for clarity (MODPO derivations & MODPO advantages)
4. We move the assumption that MODPO relies on from section 2 to section 3.1.
5. In section 3.1 (MODPO outline), we formally define the first stage of MODPO as margin reward modeling and the second stage of MODPO as language modeling.
6. In section 3.2, we rephrase the MODPO’s advantages of stability to provide more intuitions, in response to ****Reviewer GBhK’s**** question on why the multi-stage training provides benefits like stability and efficiency.
7. In section 4.1 (preliminaries), we refer readers to the newly added scaling-up experiments in the Appendix.
8. In Appendix A.3.1, we refactor the structure of this section to improve readability.
9. In Appendix E.3, we rephrase the paragraphs to provide more intuitions regarding MODPO’s stability in response to ****Reviewer GBhK’s**** question on why the multi-stage training provides benefits like stability and efficiency

---

### Meta-Review · Area_Chair_MoSe · 2023-12-07

**Metareview:**

The authors present an extension of direct preference optimization (DPO) for multiple alignment objectives, called multi-objective direct preference optimization (MODPO). Similarly to DPO, the latter is claimed to have advantages compared to a (multi-objective) reinforcement learning (from human feedback) approach. Promising empirical results in the form of Pareto sets of LMs are presented in safety alignment and long-form question answering. Advantages are also claimed with respect to computational resources.

Although the reviewers raise a number of critical points in their original reports, there is agreement that the paper has great potential. The authors showed a high level of commitment during the rebuttal phase and did their best to respond to the comments and to improve the submission. This was appreciated and positively acknowledged by all. In the discussion between authors and reviewers, some critical points could be resolved and some questions clarified. Other points remained open and were critically reconsidered in the subsequent internal discussion. The paper is still considered somewhat derivative, and certain conceptual issues and design decisions are criticised (e.g., regarding reward modeling). Moreover, the experimental part is still not found fully convincing (despite improvements made during the rebuttal phase). Eventually, the submission was found to be remain a bit behind the expectations for a top venue such as ICLR.

**Justification For Why Not Higher Score:**

Somewhat derivative, conceptual issues, experiments not fully convincing.

**Justification For Why Not Lower Score:**

N/A

---

### Decision · Program_Chairs · 2024-01-16

Reject